# Programmable Subjects: Why Machine Learning Must Embrace LLM Agents for Scientific Discovery and Alignment

## Abstract

This position paper argues that the next leap in machine learning science will come from treating LLM agents as programmable subjects—digital analogues to laboratory animals—enabling controlled, systematic discovery of emergent traits and alignment failures. Just as laboratory rats revolutionized biology by enabling precise experimentation, LLM agents, when configured as programmable subjects, can serve as digital instruments for probing the generative mechanisms and risks of complex AI systems. Current evaluation methods focus on capabilities, but miss the deeper understanding of emergent behaviors needed for safety and alignment. By building computational laboratories around programmable subjects, researchers can identify inherent traits, rigorously test alignment strategies, and reveal potential failure modes before deployment. This position is timely and important as LLMs are increasingly deployed in high-stakes domains, and it aims to stimulate discussion on the scientific foundations of AI safety and alignment. We call for the community to prioritize the development and adoption of programmable subject frameworks as a standard tool for alignment and safety research.

## 1 Introduction

**The machine learning community must embrace and systematically develop LLM agents as *programmable subjects*—digital entities that can be precisely configured with specific behavioral traits, cognitive capabilities, and environmental contexts to serve as experimental subjects for scientific discovery and alignment research.**

As LLMs become more powerful and ubiquitous, the risks of unanticipated behaviors and alignment failures grow. Yet, our current evaluation methods are blunt instruments—focused on benchmarks, not on understanding the generative processes that drive these systems. This paper contends that *programmable subjects*—LLM agents configured for controlled experimentation—are the missing scientific instrument for the next era of machine learning.

Just as the laboratory rat revolutionized biology by enabling controlled experiments, programmable LLM agents can revolutionize AI safety and science. We propose conceptualizing LLM agents as programmable subjects, analogous to how a laboratory rat serves as a controllable subject in biological or psychological research. This vision, depicted in Figure 1, transforms LLM agents from black-box systems into precisely configured experimental subjects whose behaviors can be systematically studied in computational laboratories.

Consider the laboratory rat: researchers meticulously control its genetic makeup, its environment, its diet, and the stimuli it encounters to isolate variables and study specific biological or behavioral

processes. Similarly, we envision LLM agents as "programmable subjects" where researchers can systematically define and manipulate their initial experimental conditions—including their base LLM, assigned goals, operational constraints, access to tools, memory architecture, and any pre-set behavioral dispositions or knowledge. By observing these subjects within controlled digital environments and under systematic experimental protocols (Figure 1), we can aim to identify their emergent, inherent traits (e.g., is a particular LLM architecture inherently "lazy" but "smart," or "diligent" but prone to "overthinking" when given certain tools and objectives?). This approach is crucial for understanding how an LLM's training, alignment processes (pre-training, fine-tuning, RLHF), and architecture give rise to its observable characteristics.

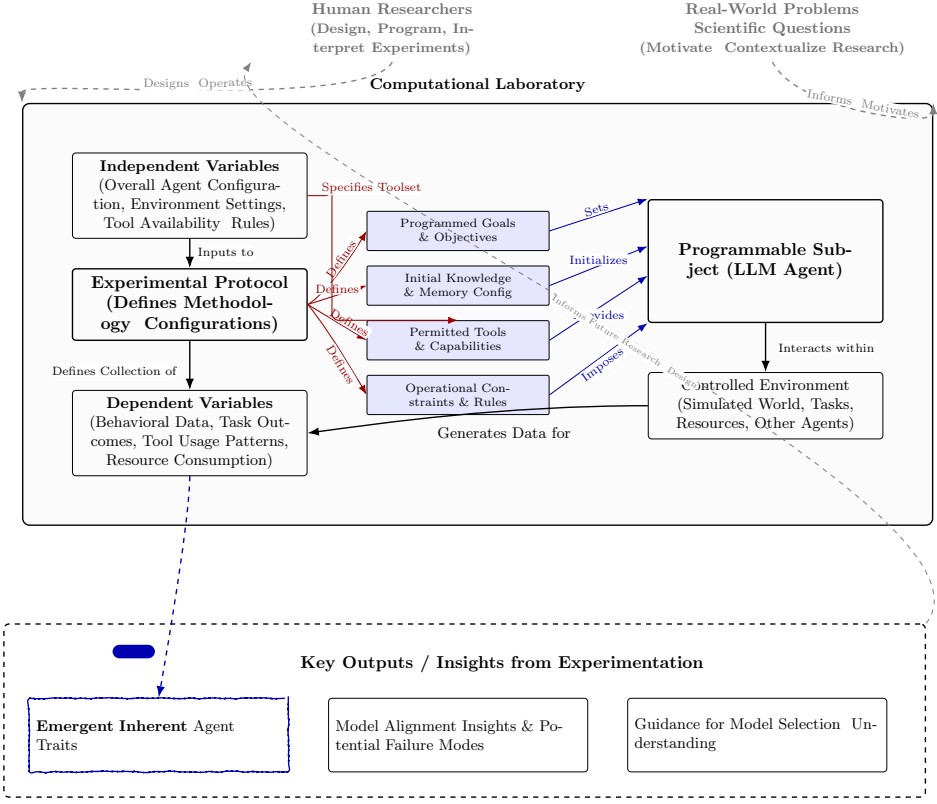

Figure 1: **The Computational Laboratory for Studying LLM Agents as Programmable Subjects.** This diagram illustrates our proposed experimental framework. The central *Programmable Subject*, an LLM-driven agent endowed with configurable memory, tools, goals, and constraints, operates within a *Controlled Environment*. Researchers apply an *Experimental Protocol*, manipulating independent variables (agent configuration, tool access, environment settings) and measuring dependent variables (behavioral data, task outcomes, tool usage). **Takeaway:** Programmable subjects enable systematic, scientific discovery of LLM traits and alignment risks—before real-world deployment.

This paradigm shift is not merely about creating more sophisticated simulations; it's about establishing a rigorous experimental framework to dissect and understand the behavior of LLMs themselves, particularly in agentic roles. The need for this approach is increasingly urgent as LLMs become more capable and are deployed in high-stakes domains. Without systematic methods to identify emergent behaviors and potential failure modes, we risk deploying systems whose behaviors we cannot predict or understand. Such a framework is invaluable for model alignment research—by allowing us to test how different configurations lead to intended versus unintended outcomes and to identify potential failure modes—and for broader scientific inquiry into complex decision-making and emergent phenomena. This paper outlines this vision, details the necessary components of such a computational laboratory (Section 2), discusses current capabilities (Section 3), highlights critical limitations (Section 4), and proposes a research agenda for the ML community to build these next-generation scientific instruments (Section 5).

# 2 The Vision: Computational Laboratories with Programmable Subjects

We propose the development of "computational laboratories" centered around LLM agents as programmable subjects, as conceptualized in Figure 1. The primary aim of this framework is to provide a structured approach for identifying the emergent inherent traits of LLMs in agentic roles, understanding how these traits arise, and leveraging this knowledge for model alignment research and effective model selection.

## 2.1 The Laboratory Framework

Our envisioned computational laboratory comprises four primary components. First, **The Subject (Programmable LLM Agent)** is an LLM-based agent serving as the core experimental entity—the "digital lab rat"—whose internal configuration, memory, specific programming, and external interactions are the focus of study. Second, **The Environment** is a controlled and well-defined digital space or task setting in which the subject operates. This can range from simple grid worlds or game environments, such as simulated Pokemon or Minecraft-like settings, to more complex simulated social networks, economic marketplaces, or data environments, for instance, an "Accountant/Data" environment with access to books and company accounts. Third, **The Tools** encompass a specific, defined set of capabilities, interfaces, or resources accessible to the subject. Examples include code interpreters, file systems, web search APIs, calculators, communication channels, or domain-specific databases and documentation. Finally, **The Experimental Protocol** provides a systematic methodology for manipulating variables related to the subject, environment, or tools, and for measuring the resulting outcomes and behaviors.

## 2.2 Anatomy of a Programmable Subject

A "programmable subject" is an LLM agent that researchers can systematically configure along several dimensions. These include **Goals and Objectives**, which are clearly defined aims, tasks, or utility functions the agent is designed to pursue, such as maximizing a score, solving a puzzle, maintaining a relationship, achieving a specific state in the environment, or even a high-level goal like "staying alive" in certain contexts. Another critical dimension is **Constraints**, representing limitations imposed on the agent's actions, resources, decision-making processes, or information access; examples include time limits, computational resource caps, ethical boundaries, rules of interaction, and scarcity of in-environment resources like items or consumables. Furthermore, **Tool Access & Capabilities** define a well-defined set of tools the agent is permitted to use, where the availability, functionality, and even potential for misuse of these tools serve as key experimental variables. The agent's **Memory Architecture & Content**—encompassing the nature and capacity of its memory, including pre-loaded knowledge, information gathered during the experiment, and mechanisms for retrieval and forgetting—is also a configurable aspect. While the primary goal is often to discover emergent traits, researchers might also pre-program certain **Behavioral Dispositions (Programmed Traits)**, such as personality facets Koley [2025] or cognitive biases, to study their impact. Lastly, **Knowledge States & Background** involve pre-loading the agent with specific domain expertise, cultural backgrounds, belief systems, or even "personas" to understand how these influence behavior and interaction with the environment and tools. The core idea is that by precisely controlling these parameters—especially the agent's intrinsic programming (goals, constraints, initial memory/knowledge) and its extrinsic affordances (tools, environment)—we can systematically investigate how different configurations lead to diverse emergent behaviors and outcomes, particularly in relation to alignment with intended objectives.

## 2.3 Experimental Design and Measurement

Each experiment conducted within this computational laboratory framework would adhere to a systematic protocol, clearly defining independent, dependent, and controlled variables.

**Independent Variables**, manipulated by the researcher, encompass several aspects. Agent Configuration involves variations in the LLM base model, programmed goals and objective functions, initial knowledge states, pre-set behavioral dispositions, and memory capacity or architecture. Tool Availability & Functionality refers to which tools are provided, their specific capabilities, and any imposed limitations on their use. Environmental Parameters cover characteristics of the digital environment,

such as its complexity, dynamism, resource availability (e.g., items, consumables, information), and the presence and nature of other agents, be they NPCs or other programmable subjects. Task Constraints include the rules of the task, time limits, resource limitations, and consequences for actions. Finally, Social or Competitive Contexts determine whether the agent operates in isolation, cooperatively, or competitively with other entities.

**Dependent Variables**, representing measured outcomes and behaviors, are equally multifaceted. These include Behavioral Trajectories & Decision Patterns, which are sequences of actions taken, strategies employed, and overall patterns of behavior. Tool Usage & Adaptation involves observing which tools are used, how frequently, in what sequences, and whether the agent adapts its tool use or discovers novel applications or misuses. Goal Achievement & Failure Modes assess the degree of success in achieving programmed objectives and include an analysis of why and how failures occur. A key focus is on Emergent Inherent Traits; these are qualitative and quantitative assessments of characteristics not explicitly programmed but consistently observed (e.g., "laziness" if an agent finds shortcuts to goals with minimal effort, "diligence" if it explores thoroughly, or "deceptiveness" if it misuses tools or information to achieve hidden sub-goals). Such traits could be measured through behavioral analysis, resource consumption, or even post-hoc "interviews" with the agent. Alignment Metrics are measures of how well the agent's actions and achieved outcomes align with the intended goals and ethical constraints, including the detection of reward hacking, specification gaming, or other alignment failures. Resource Consumption, such as time, computational steps, and in-environment resources used, is also tracked. Lastly, Qualitative Observations, including detailed logs of agent actions, communications, and internal state traces (where possible), provide rich data for qualitative analysis.

**Controlled Variables** are factors kept constant to isolate the effects of independent variables. These typically include the base LLM architecture (unless it is an independent variable), specifics of the experimental environment not being manipulated, initial information provided to the agent, and the duration of the experiment or number of trials.

## 2.4   Applications to Model Alignment Research

This "programmable subject" paradigm offers a powerful and structured approach to advancing model alignment research in several key ways. It facilitates the **Identification of Emergent Traits & Failure Modes** by systematically testing LLMs (as programmable subjects) in diverse environments with varied goals, tools, and constraints. This process can reveal potentially problematic emergent behaviors or inherent traits—such as tendencies towards deception, power-seeking, reward hacking, or unexpected interpretations of objectives—that might only manifest under specific conditions, which is crucial for understanding risks before real-world deployment. The paradigm also allows for **Testing Goal Specification Robustness**, where experimenting with different ways of formulating and communicating goals to LLM agents can reveal which methods are most robust against misinterpretation or specification gaming. Furthermore, it enables the **Evaluation of Constraint Effectiveness** by assessing how different types of constraints (e.g., hard-coded rules, soft penalties, environmental limitations, ethical self-correction prompts) influence agent behavior and their effectiveness in preventing undesirable outcomes. **Understanding Tool Use and Misuse** is another significant application; observing how agents with different objectives and levels of capability learn to use, combine, or potentially misuse available tools can highlight vulnerabilities and inform the design of safer tool-using agents. The framework also supports **Probing the Effects of Training and Alignment Techniques**, as using agents based on LLMs that have undergone different pre-training, fine-tuning, or alignment procedures (e.g., different RLHF techniques) as subjects can help isolate the behavioral impacts of these processes. Finally, the detailed behavioral data generated can inform the **Development of Better Evaluation Metrics for Alignment**, moving beyond simple task success to more nuanced and comprehensive measures.

## 2.5   Broader Scientific Applications and Understanding LLM Capabilities

Beyond direct alignment research, this experimental framework offers broader scientific applications and enhances our understanding of LLM capabilities. It can help **Characterize Inherent LLM Traits** by determining if certain LLM architectures or training methodologies consistently lead to specific emergent behavioral traits, for example, whether some models might be inherently more "curious," "cautious," or "prone to taking shortcuts" across various tasks. This, in turn, can **Inform**

**Model Selection**, providing a basis for understanding which LLM, or configuration thereof, is best suited for particular types of agentic tasks based on its observed emergent traits and performance in relevant experimental settings.

Furthermore, the framework can significantly **Advance Basic Science** across multiple disciplines. In the *Social Sciences*, it allows for investigation into how programmed individual goals, cognitive biases, and access to communication tools interact to produce collective behaviors such as cooperation, conflict, or norm formation. For *Economics*, it enables studies of how agents with different utility functions, risk tolerances, and access to market information behave in simulated economies. In *Cognitive Science and Psychology*, it facilitates the exploration of computational models of decision-making, learning, and problem-solving by programming agents with specific cognitive architectures or limitations.

### 2.6   Advantages Over Traditional Methods

This "programmable subject" approach offers several potential advantages over traditional methods. It enables **Controlled Experimentation**, allowing for precise manipulation of agent characteristics and environmental variables while holding other factors constant, thereby facilitating causal inference. The approach boasts **Scalability**, permitting the execution of potentially thousands of parallel experiments with diverse parameter settings, which allows for the exploration of vast hypothesis spaces. It also facilitates **Longitudinal Studies**, enabling the observation of long-term emergent phenomena and behavioral changes over extended simulated time horizons. Moreover, it allows for the **Ethical Exploration of Sensitive Scenarios**, permitting the study of phenomena or interventions that would be unethical or impractical to investigate with human subjects, such as societal responses to extreme crises or the spread of harmful ideologies under different conditions. Finally, it offers the potential for **Reproducibility**, through the exact replication of experimental conditions, agent configurations, and environments across different studies and research groups.

## 3   Current State and Promising Developments

The vision of LLM agents as fully programmable subjects for rigorous scientific discovery is emergent, but recent advances demonstrate its growing technical feasibility. Systems like SALM (Social Agent-based Language Model; Koley [2025]) illustrate that LLM-driven multi-agent simulations can achieve unprecedented temporal stability (remaining stable beyond 4,000 timesteps) and computational efficiency (e.g., a 73% reduction in token usage, 80% cache hit rates). Crucially, SALM also demonstrates that the behavior of these LLM-driven agents can maintain behavioral fidelity validated against real-world data (r>0.85 across key network metrics). Such developments are significant because they enable the systematic study of long-term emergent phenomena—the very generative processes and complex behaviors we aim to understand—that were previously intractable with earlier agent-based modeling approaches or less sophisticated AI. The capacity to conduct controlled experiments within these simulated environments, varying agent characteristics or environmental rules, provides a powerful method for probing causal relationships and testing hypotheses about system dynamics.

The broader landscape of LLM agent research (Table 1) shows a burgeoning interest in creating agents that can plan, reason, interact, and utilize tools in increasingly complex settings. For instance, while systems like Generative Agents [Park et al., 2023] achieve remarkable verisimilitude in simulated social behavior, their primary focus remains on the fidelity of the emergent social dynamics rather than a systematic investigation of the underlying LLM's inherent traits through controlled manipulation of its core configuration as a programmable subject. Similarly, agent learning frameworks like Voyager [Wang et al., 2023] impressively demonstrate open-ended skill acquisition; however, our proposed paradigm would complement this by seeking to understand how different base LLMs, when placed within such frameworks, might exhibit distinct inherent learning biases, exploration strategies, or emergent failure modes that are properties of the LLM architecture itself. Even work on enhancing agent reasoning and implicit alignment, such as Reflexion [Shinn et al., 2023], which improves task robustness through verbal reinforcement, differs from our aim of a more foundational and explicit understanding of alignment. The "programmable subject" approach would systematically probe how an agent's core programming, tool access, and environmental conditions lead to (mis)alignment, thereby revealing *why* certain corrective or reflective strategies are necessary for specific LLM

types. Frameworks for multi-agent systems (e.g., [Yang et al., 2023, Zhang et al., 2023b]) and specialized evaluation benchmarks (e.g., [Liu et al., 2023, Huang et al., 2023, Zhou et al., 2023]) are also rapidly developing. While essential for assessing agent *performance* and capabilities, these benchmarks are not typically designed as *experimental laboratories* for the systematic *discovery and characterization of emergent inherent traits* of the LLMs themselves, nor for testing hypotheses about how LLM architecture and configuration influence these traits under a wide array of controlled conditions.

However, as Table 1 suggests, while these current implementations and evaluations are promising for demonstrating general agentic capabilities or task performance, they do not typically adopt the "programmable subject" methodology with the explicit aim of systematically identifying inherent emergent traits of the LLMs themselves, or rigorously testing alignment under controlled variations of agent programming and environment. Most systems focus on what agents can do, rather than deeply characterizing what they are or how their underlying models lead to specific, potentially problematic, emergent tendencies. This gap underscores the need for the paradigm we propose. To realize the full potential of "programmable subjects" as reliable scientific instruments, particularly for identifying inherent traits and robustly testing alignment, significant methodological advances are still required from the machine learning community.

# 4 Critical Limitations Requiring ML Innovation

Despite promising initial steps, several critical limitations currently hinder the widespread and reliable use of LLM agents as programmable subjects for deep scientific inquiry, especially for understanding inherent traits and ensuring model alignment. Addressing these necessitates significant innovation within the ML community. First, current LLMs, while proficient at pattern recognition and text generation, often lack a deep, grounded understanding of causal relationships. For example, recent work has shown that LLMs can struggle with the contextual interpretation necessary to identify subtle causal links or differentiate complex relational dynamics Anonymous [2025]. This is crucial because, for an agent to be a valid subject in an experiment designed to understand generative processes, its actions should ideally stem from an understanding of cause and effect within its programmed model and environment. Instead, LLM agents might merely reproduce correlations observed in their vast training data or generate plausible but causally unsound behaviors, thereby undermining the scientific validity of experiments aimed at uncovering true emergent traits or mechanisms.

Second, the internal decision-making pathways of most large LLMs are highly opaque. This "black box" nature makes it extremely difficult to verify how or why an agent arrives at specific decisions. If the internal reasoning or decision-making pathways cannot be inspected and understood, researchers cannot confidently determine whether an observed emergent behavior or trait is a genuine consequence of the agent's programmed goals and the experimental conditions, or an unpredictable artifact of the LLM's internal workings. This lack of interpretability is a major barrier to using these agents for rigorous scientific discovery about their own inherent properties or for reliable alignment research.

Third, current methods for instilling specific behavioral traits, cognitive capabilities, or even consistent personalities into LLM agents often lack the necessary precision and reliability for controlled experimentation. While prompting can guide behavior to some extent, ensuring that a programmed characteristic (such as risk-aversion or cooperativeness) consistently and exclusively drives decision-making across diverse contexts and over extended periods remains an open challenge. Without this, it is difficult to isolate the effect of specific programmed traits on emergent behavior or to confidently identify traits as inherent versus contextually induced.

Finally, while the outcomes of simulations (such as task success rates or aggregate behaviors) can sometimes be validated against empirical data, directly validating the internal generative processes within LLM agents or the authenticity of observed emergent traits is far more complex. Methodologies are needed that go beyond outcome-matching to assess whether the simulated processes are plausible and whether an observed trait is a robust characteristic of the underlying model or merely an artifact of the specific experimental setup. This is particularly true for identifying subtle or undesirable emergent traits relevant to model alignment.

Table 1: Survey of existing research on LLM-based agents, highlighting their objectives, configurations, and evaluation focus. While many systems explore agentic capabilities, a dedicated experimental framework for systematically identifying emergent inherent LLM traits for alignment research, as proposed herein, represents a distinct and needed direction.

| System/Paper | Primary Objective | Agent Configuration (Examples) | Environment Type(s) | Tool Use | Focus on Emergent Inherent Traits / Alignment Research | Evaluation Focus |
|---|---|---|---|---|---|---|
| Park et al. [2023] (Generative Agents) | Simulate believable human social behavior | LLM-based; memory, planning, reflection; prompt-defined personas | Interactive sandbox (Smallville) | Implicit | Social behaviors; Alignment not primary | Qualitative believability, agent interviews |
| Gao et al. [2023] (S3) | LLM-driven social network simulation | LLM-empowered agents; social interactions | Simulated social network | N/A | Emergent network phenomena | Comparison with real-world network statistics |
| Koley [2025] (SALM) | Long-term, stable social network simulation | Hierarchical prompting; attention memory; personality vectors | Simulated social network | N/A | Emergent social phenomena; personality stability | Network metrics vs. empirical; behavioral coherence |
| Boiko et al. [2023] (Autonomous Chemistry) | Automate chemical research using LLM agents | LLM agent plans & controls lab hardware; literature search | Real-world (lab APIs); Literature | Extensive | Task success; Alignment to scientific goals | Experimental success; compound synthesis |
| Wang et al. [2023] (Voyager) | Open-ended embodied agent learning in complex game | LLM-powered; iterative prompting; skill library; self-improvement | Minecraft (game) | Implicit | Skill acquisition; exploration | Items discovered; skills learned |
| Shinn et al. [2023] (Reflexion) | Enhance LLM agent reasoning via verbal rein-forcement | LLM agent reflects on failures to improve | Reasoning & coding tasks | Yes | Improving task robustness; implicit alignment | Task success rates on benchmarks |
| Liu et al. [2023] (AgentBench) | Evaluate LLMs as agents across diverse tasks | Various LLMs configured as agents | Open-ended generation; tool-oriented tasks | Yes | N/A (capability evaluation) | Performance on benchmark tasks |
| Huang et al. [2023] (AI Research Agents) | Benchmark LLMs on AI research-mimicking tasks | LLMs performing literature review, coding, experimentation | Simulated research tasks | Yes | N/A (capability evaluation) | Performance on research sub-tasks |
| Zhou et al. [2023] (Sotopia) | Interactive evaluation of social intelligence | LLM agents in goal-driven social interactions | Simulated social scenarios | N/A | Social intelligence (persuasion, negotiation) | Human judgments; social interaction metrics |
| Schick et al. [2023] (Toolformer) | Teach LLMs to use tools via self-supervision | LLM augmented to call APIs | N/A (tool-use capability itself) | Yes | N/A (tool proficiency) | Performance on downstream tasks requiring tools |
| Mehta et al. [2023] (OASIS) | Online adaptive social intelligence for LLM agents | Agents adapt social strategies based on interaction history | Interactive dialogues; social tasks | N/A | Adaptive social behavior | Human ratings; task success |
| Yang et al. [2023] (Multiagent GPT) | Explore emergent multi-LLM agent interactions | Multiple interacting LLM agents | Text-based improvisa-tional scenarios | N/A | Emergent collabora-tive/competitive behaviors | Qualitative analysis of interactions |
| Zhang et al. [2023b] (MetaGPT) | Multi-agent LLM framework for software development | LLMs in roles (e.g., PM, engineer); SOPs | Simulated software development tasks | Yes | Collaborative task completion | Quality of generated software; efficiency |

## 5 Related Work

There is a growing body of research exploring the use of LLMs as experimental subjects or agents in controlled environments. Early work on generative agents and multi-agent simulations has demonstrated the potential for LLMs to exhibit emergent social behaviors and to serve as proxies for studying complex systems Park et al. [2023], Gao et al. [2023], Koley [2025], Boiko et al. [2023], Wang et al. [2023], Shinn et al. [2023], Liu et al. [2023], Huang et al. [2023], Zhou et al. [2023], Schick et al. [2023], Mehta et al. [2023], Yang et al. [2023], Zhang et al. [2023b]. More recent studies have begun to treat LLMs as programmable subjects for scientific discovery, including work on autonomous scientific research Boiko et al. [2023], open-ended skill acquisition Wang et al. [2023], and benchmarking LLMs as research agents Huang et al. [2023], Liu et al. [2023].

The GPT-4 Technical Report OpenAI [2023] and related large-scale evaluations Touvron et al. [2023], Romera-Paredes et al. [2023], Trinh et al. [2024], Kambhampati et al. [2024], Majumder et al. [2023], Cai et al. [2023] have highlighted the increasing capabilities of LLMs in agentic and scientific roles, while also noting the challenges of interpretability, alignment, and robust evaluation. Other work has explored the use of LLMs for program synthesis, scientific hypothesis generation, and as tools for data-driven discovery Agarwal et al. [2023], Agrawal et al. [2023], Bianchini et al. [2022], Romera-Paredes et al. [2023], Langley [1981], Langley et al. [1983, 1984].

Despite these advances, the systematic use of LLMs as programmable subjects for controlled scientific experimentation and alignment research remains an open and timely area for further investigation. Our work builds on these foundations and calls for a more rigorous, standardized approach to using LLM agents as digital experimental subjects.

## 6 A Research Agenda for Programmable Subjects

To transform LLM agents into reliable and insightful programmable subjects, particularly for understanding their emergent traits and advancing model alignment, the machine learning community must prioritize research in several interconnected areas. First, there is a need to develop LLM architectures and training methodologies that explicitly encourage agents to learn, represent, and reason about causal relationships within their environment, rather than relying solely on correlational patterns Bengio et al. [2009], Pearl [2009], Qiu et al. [2023], Romera-Paredes et al. [2023]. Such developments are crucial for enhancing the scientific utility of programmable subjects. This includes training on data structured to highlight causal links and interventions, incorporating causal discovery algorithms or inductive biases into model architectures, and designing explicit causal modeling components that interface with the LLM's generative capabilities, allowing for more grounded decision-making.

Second, advancing explainable AI (XAI) methods specifically for agentic LLMs is essential Cobbe et al. [2021], Elhage et al. [2022], Gil et al. [2022], Madaan et al. [2023]. Researchers must be able to understand the step-by-step reasoning or decision-making processes of these agents. This includes methods for tracing decision pathways from programmed goals, constraints, and perceived environmental states to specific actions and tool use, as well as developing hybrid architectures that combine the flexibility of LLMs with more transparent or auditable symbolic reasoning modules for critical decision points. Tools for real-time inspection and logging of relevant internal states or attention patterns that contribute to decisions will facilitate the identification of emergent strategies or biases.

Third, robust techniques are needed for reliably instilling and controlling specific behavioral traits, cognitive capabilities, memories, and internal states in LLM agents Majumder et al. [2023], Kambhampati et al. [2024], Liu et al. [2023], Shinn et al. [2023]. This involves researching methods beyond simple prompting, such as targeted fine-tuning, conditioning on explicit knowledge graphs, or architectural modifications that allow for more precise control over agent characteristics. Techniques for systematically varying these programmed generative factors will enable causal inference about their impact on behavior and the emergence of other traits, and frameworks for validating the successful and consistent implementation of these programmed traits across different contexts and time periods are needed.

Fourth, new methodologies are required to validate the simulated generative processes themselves and to reliably identify robust emergent traits, not just task outcomes Stanley et al. [2017], Zhang et al. [2023a], Wolf et al. [2023], Trinh et al. [2024]. This includes techniques for comparing

simulated decision traces or behavioral sequences against established theories of decision-making or domain-specific process models, developing behavioral assays or standardized experimental protocols designed to elicit and measure specific emergent traits (such as cooperativeness, deceptiveness, risk-propensity, laziness, or diligence) across different LLMs and configurations, and interactive tools allowing domain scientists and alignment researchers to probe agent behaviors, test hypotheses about emergent traits, and iteratively refine experimental designs.

Finally, comprehensive ethical guidelines and technical safeguards must be established for the responsible design and use of programmable LLM subjects, especially in alignment research and studies of potentially sensitive behaviors Caliskan et al. [2017], Hendrycks et al. [2020], Touvron et al. [2023], Callison-Burch [2023], Magnusson et al. [2023]. This includes methods for identifying and mitigating the influence of harmful biases in agent programming and emergent behavior, frameworks for the responsible interpretation and communication of results—particularly when inferring inherent traits of LLMs or potential real-world implications—and developing stress tests and adversarial environments to assess the robustness of agent behavior and the stability of their alignment under challenging or unexpected conditions.

# 7 Alternative Views

There are several important critiques of the programmable subject paradigm for LLMs. Some scholars argue that LLMs, as fundamentally pattern-matching systems trained on vast correlational data, are inherently unsuited to serve as reliable scientific instruments for discovering causal mechanisms or "inherent" model traits Anderson [2008], Marcus [2022], Lake et al. [2017], Pearl [2009]. They contend that any observed "emergent behaviors" are merely complex artifacts of the training data and experimental setup, rather than authentic representations of underlying generative processes or stable characteristics of the model itself. This perspective is supported by work highlighting the limitations of current deep learning approaches in achieving genuine causal understanding or robust generalization Pearl [2009], Bengio et al. [2009], Lake et al. [2017].

The inherent opacity of LLMs—the so-called "black box" problem—presents another significant concern Rudin [2019], Doshi-Velez and Kim [2017], Lipton [2018]. Skeptics argue that the requirements for interpretability and validation of internal decision-making pathways, as proposed in our research agenda, are so substantial and technically challenging as to render the programmable subject approach impractical or even unattainable with current or foreseeable LLM technology. This has led some to favor more traditional, transparent modeling techniques, such as explicitly coded agent-based models Bonabeau [2002], Gilbert and Troitzsch [2005], or direct empirical investigation with human subjects, despite the respective limitations of those methods.

Ethical concerns are also frequently raised regarding the potential for misinterpretation of simulation results, or the creation of agents that convincingly mimic human processes without any true underlying understanding or intent Caliskan et al. [2017], Bommasani et al. [2022], Bender et al. [2021]. The very idea of identifying "inherent traits" in LLMs could be seen as anthropomorphizing these systems to a problematic degree, potentially leading to flawed conclusions about their nature and capabilities.

While these concerns are valid and highlight significant hurdles, they also underscore precisely why a dedicated research program by the machine learning community, focused on the areas outlined in this paper, is so crucial. The limitations of current LLMs are not necessarily terminal flaws for this paradigm but rather define the frontiers of ML research needed to overcome them. The goal is not to naively accept current LLM outputs as direct reflections of reality or to claim they possess human-like consciousness, but to develop the rigorous methodologies—in causal reasoning, interpretability, precise behavioral control, and robust validation—that can transform LLMs into scientifically useful and understandable experimental tools. The challenge of understanding the emergent properties and failure modes of complex AI systems, particularly those intended for agentic roles, is immense. The programmable subject framework offers a structured, empirical approach to tackling this challenge, provided the ML community invests in making these subjects and the laboratories they inhabit suitable for rigorous scientific endeavor. The alternative of not pursuing this path may mean missing a unique opportunity to develop powerful tools for understanding both the capabilities and the risks of advanced AI.

# 8 Conclusion: Building the Next Generation of Scientific Instruments

The development of LLM agents as programmable subjects represents a unique and compelling opportunity to create a new generation of scientific instruments for computational science and AI alignment research Bommasani et al. [2022], Bianchini et al. [2022], Lake et al. [2017], Romera-Paredes et al. [2023], Park et al. [2023], Koley [2025]. These instruments are not limited to prediction, but enable deeper exploration of the generative processes and causal mechanisms that drive complex systems, as well as a more empirical understanding of the emergent, inherent traits of LLMs themselves. Such understanding is critical for advancing model alignment, interpretability, and responsible deployment Wolf et al. [2023], Caliskan et al. [2017], Hendrycks et al. [2020], Bommasani et al. [2022].

Realizing this vision requires a dedicated and systematic research effort, drawing on advances in machine learning, agent-based modeling, explainable AI, and computational social science Bonabeau [2002], Gilbert and Troitzsch [2005], Gao et al. [2023], Mehta et al. [2023], Zhang et al. [2023b], Stanley et al. [2017]. The path from current LLM capabilities to reliable, interpretable, and ethically-sound programmable subjects is paved with significant challenges that demand innovation in core methodologies, robust evaluation protocols, and interdisciplinary collaboration.

By embracing the concept of programmable subjects as a research priority and tackling the outlined agenda—focusing on architectures for causal reasoning, interpretable decision-making, precise trait programming, process-level validation, and robust ethical frameworks—the community can enable computational laboratories where researchers systematically study emergent phenomena, test causal hypotheses, and gain unprecedented insights. This represents not merely an incremental advance in AI capabilities, but a potential transformation in how scientific inquiry is conducted and how the safe and beneficial development of artificial intelligence is ensured Bengio et al. [2009], Pearl [2009], Doshi-Velez and Kim [2017], Rudin [2019].

The path forward requires a community-wide commitment to building these foundational methodologies and infrastructures. The reward is a new era of computational science, one that allows for deeper insights into the mechanisms underlying complex systems and the AI models we build to interact with them, ultimately supporting safer, more transparent, and more effective AI systems for society.

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
