# OpenReview forum: "A Call for LLMs Agents as Programmable Subjects"
_NeurIPS.cc/2025/Position_Paper_Track — Submitted to NeurIPS 2025 Position Paper Track_

### Official Review · Reviewer_qceQ · 2025-08-04

**Significance:** 2
**Presentation:** 2
**Rating:** 4
**Confidence:** 3

**Summary:**

The authors argue that ML should treat LLM agents as ``programmable subjects'' (digital analogs of lab animals) studied in virtual laboratories with controlled environments, tools, and protocols to elicit emergent traits and expose alignment failures before deployment. It then proposes a research agenda (causal reasoning, interpretability, precise trait programming, process-level validation) and notes current limits (weak causal understanding, opacity, and unstable trait control) that this position is meant to tackle.

**Strengths:**

1. The framework reads pretty actionable. Instead of a loose metaphor, the authors detailed the controllable variables (goals, tools, memory, and constraints) and measurable outputs.

2. It explicitly surveys counterarguments (causality gaps, opacity, anthropomorphism) and positions them as research targets, motivating agendas in causal reasoning, interpretability, reliable trait programming, and validation.

**Weaknesses:**

1. The draft shows grammar and typography/layout problems (e.g., truncated text in the Tool Use column and overlapping, unevenly spaced labels in at least one figure), which detract from readability and polish.

2. The paper’s own related-work section cites several efforts that already adopt controlled, lab-like evaluations of LLM agents. That makes the central framing feel incremental rather than new. Could you clarify what is substantively novel? A side-by-side table contrasting your paradigm against the closest baselines would help.

3. The text lists caveats, but the leap to “therefore we need a dedicated program” is asserted rather than argued. You might consider making the causal chain explicit: (i) state each obstacle, (ii) show why current eval paradigms cannot resolve it,(iii) specify the new methods your framework uniquely contributes, (iv) define success criteria and milestones.

4. To make the vision concrete and assessable, please add a concise real-world case study where this “computational laboratory” would yield nontrivial insight. This will let readers evaluate feasibility, falsifiability, and the incremental value of your paradigm over existing evaluation setups.

**Questions:**

1. What, beyond “LLMs as drop-in agents,” distinguishes your framework from conventional ABM in design, mechanisms, and outcomes?

2. How would you situate LLM-based open-ended simulation in your discussion? One example is [1].

**References:**
[1] Lai, S., Potter, Y., Kim, J., Zhuang, R., Song, D., & Evans, J. (2024, July). Position: Evolving AI collectives enhance human diversity and enable self-regulation. In Forty-first International Conference on Machine Learning.

**Alternative Position:**

Yes, and alternative positions are well-considered and addressed by the argument

**Author Identification:**

No.

**Context:**

2

**Discussion:**

3

**Ethics:**

["NO or VERY MINOR ethics concerns only"]

**Position:**

Yes, the paper argues for or against a position related to machine learning.

**Support:**

3

**Thoroughness:**

4

---

### Official Review · Reviewer_PU5v · 2025-08-09

**Significance:** 2
**Presentation:** 1
**Rating:** 2
**Confidence:** 3

**Summary:**

There has been great progress/interest in LLM agents. The paper argues that these agents should be treated as “programmable subjects, similar to laboratory animals in science. This allows to  systematically study emergent traits, alignment failures, and safe deployment strategies. Basically, this can allow for appropriate controlled experimental setup. The authors propose a lab and programmable subject paradigms. The paper then focuses on alignment research as an application.

**Strengths:**

- The paper bring a new perspective to LLM agents.

- The paper emphasizes on interpretability of emergent behaviours of LLMs Which is very relevant to the community.

- The proposed paradigm with different variables and programmable subjects seems intuitive.

**Weaknesses:**

- The paper violates the official format. (MAJOR) They have used an extra page and page 7 is incorrectly formatted, table bleeds out to the margin. (Minor) Additionally the open review title is not the same as the paper titles.

- At some times, the papers objective seems confusing, where the authors interchange between.

## (I am giving a rating of 2 because the paper did not respect the page limit.)

**Questions:**

- Figure 1 is confusing. Not sure where does the informs motivates arrow points towards. Some of the test is overlayed on the other, i would appreciate if there exists no overlay on top of each other to be able to read every word well.

- How will you operationally define and validate an “inherent trait” so it’s not just a contextual behavior?

- Who do you think should control access to such programmable subject platforms? Given there exists many stakeholders?

**Alternative Position:**

Yes, and alternative positions are well-considered and addressed by the argument

**Author Identification:**

No.

**Context:**

2

**Discussion:**

3

**Ethics:**

["NO or VERY MINOR ethics concerns only"]

**Position:**

Yes, the paper argues for or against a position related to machine learning.

**Support:**

2

**Thoroughness:**

3

---

### Official Review · Reviewer_WYze · 2025-08-13

**Significance:** 3
**Presentation:** 4
**Rating:** 5
**Confidence:** 4

**Summary:**

This paper proposes treating LLM agents as programmable subjects (similar to digital laboratory animals), allowing researchers to systematically configure, constrain, and study their behaviors in controlled computational environments.

**Strengths:**

Comprehensive review of papers framing LLM agents as a standardized experimental instrument for scientific discovery (Table 1)

**Weaknesses:**

There have been a large body of literature using LLMs for (social) simulations. It might be helpful if the authors could further demonstrate the novelty of their position.

**Questions:**

- Do the authors envision the programmable subject paradigm as being limited to predicting the behaviors of multi-agent AI systems, or could it also be applied in human–AI interaction contexts?  (e.g., using LLMs as simulators of human participants (c.f. Harding et al., 2024, AI & Society, 39(5), 2603–2605)
- If the latter, in what specific scenarios (e.g., prototyping conversational interfaces, simulating negotiation partners, testing safety guardrails in mixed human-AI agent teams) would this approach be most reliable and informative?

**Alternative Position:**

Yes, and alternative positions are well-considered and addressed by the argument

**Author Identification:**

No.

**Context:**

4

**Discussion:**

3

**Ethics:**

["NO or VERY MINOR ethics concerns only"]

**Position:**

Yes, the paper argues for or against a position related to machine learning.

**Support:**

3

**Thoroughness:**

4

---

### Meta-Review · Area_Chair_Usz3 · 2025-09-18

**Rating:** 4
**Confidence:** 4

**Strengths:**

The paper presents a compelling framing and a good guiding metaphor (LLMs as programmable subjects akin to laboratory subjects). The literature review is comprehensive, including Table 1's survey of existing research on LLM agents. The paper also did a great job addressing potential counter-arguments.

**Weaknesses:**

There were major formatting issues, including extra page and tables that bleed over.

Reviewers also expressed concern that the current work isn't sufficiently distinguished from existing work. In particular, it would help to compare with agent based models.

Reviewers also felt the paper could be improved by a compelling case study to make it more concrete.

**Questions:**

How would you operationally define and validate an "inherent trait" to ensure that it is not just contextual behavior? What would be the standard of evidence for innateness?

**Ethics:**

None mentioned

**Thoroughness:**

4

---

### Decision · Program_Chairs · 2025-09-26

Reject